# Parametric Simplex Method for Sparse Learning

**Haotian Pang**[‡]    **Robert Vanderbei**[‡]    **Han Liu**[⋆‡]    **Tuo Zhao**[◇]
[‡]Princeton University  [⋆]Tencent AI Lab  [‡]Northwestern University  [◇]Georgia Tech[*]

## Abstract

High dimensional sparse learning has imposed a great computational challenge to large scale data analysis. In this paper, we are interested in a broad class of sparse learning approaches formulated as linear programs parametrized by a *regularization factor*, and solve them by the parametric simplex method (PSM). Our parametric simplex method offers significant advantages over other competing methods: (1) PSM naturally obtains the complete solution path for all values of the regularization parameter; (2) PSM provides a high precision dual certificate stopping criterion; (3) PSM yields sparse solutions through very few iterations, and the solution sparsity significantly reduces the computational cost per iteration. Particularly, we demonstrate the superiority of PSM over various sparse learning approaches, including Dantzig selector for sparse linear regression, LAD-Lasso for sparse robust linear regression, CLIME for sparse precision matrix estimation, sparse differential network estimation, and sparse Linear Programming Discriminant (LPD) analysis. We then provide sufficient conditions under which PSM always outputs sparse solutions such that its computational performance can be significantly boosted. Thorough numerical experiments are provided to demonstrate the outstanding performance of the PSM method.

## 1  Introduction

A broad class of sparse learning approaches can be formulated as high dimensional optimization problems. A well known example is Dantzig Selector, which minimizes a sparsity-inducing $\ell_1$ norm with an $\ell_\infty$ norm constraint. Specifically, let $X \in \mathbb{R}^{n \times d}$ be a design matrix, $y \in \mathbb{R}^n$ be a response vector, and $\theta \in \mathbb{R}^d$ be the model parameter. Dantzig Selector can be formulated as the solution to the following convex program,

$$\widehat{\theta} = \underset{\theta}{\operatorname{argmin}} \ \|\theta\|_1 \quad \text{s.t.} \ \|X^\top(y - X\theta)\|_\infty \leq \lambda. \tag{1.1}$$

where $\|\cdot\|_\infty$ and $\|\cdot\|_1$ denote the $\ell_\infty$ and $\ell_1$ norms respectively, and $\lambda > 0$ is a regularization factor. Candes and Tao (2007) suggest to rewrite (1.1) as a linear program solved by linear program solvers.

Dantzig Selector motivates many other sparse learning approaches, which also apply a regularization factor to tune the desired solution. Many of them can be written as a linear program in the following generic form with either equality constraints:

$$\max_x (c + \lambda \bar{c})^\top x \quad \text{s.t.} \ Ax = b + \lambda \bar{b}, \ x \geq \mathbf{0}, \tag{1.2}$$

or inequality constraints:

$$\max_x (c + \lambda \bar{c})^\top x \quad \text{s.t.} \ Ax \leq b + \lambda \bar{b}, \ x \geq \mathbf{0}. \tag{1.3}$$

Existing literature usually suggests the popular interior point method (IPM) to solve (1.2) and (1.3). The interior point method is famous for solving linear programs in polynomial time. Specifically, the interior point method uses the log barrier to handle the constraints, and rewrites (1.2) or (1.3)

---

[*]Correspondence to Tuo Zhao: `tuo.zhao@isye.gatech.edu`.

as a unconstrained program, which is further solved by the Newton's method. Since the log barrier requires the Newton's method to only iterate within the interior of the feasible region, IPM cannot yield exact sparse iterates, and cannot take advantage of sparsity to boost the computation.

An alternative approach is the simplex method. From a geometric perspective, the classical simplex method iterates over the vertices of a polytope. Algebraically, the algorithm involves moving from one partition of the basic and nonbasic variables to another. Each partition deviates from the previous in that one basic variable gets swapped with one nonbasic variable in a process called *pivoting*. Different variants of the simplex method are defined by different rules of pivoting. The simplex method has been shown to work well in practice, even though its worst-case iteration complexity has been shown to scale exponentially with the problem scale in existing literature.

More recently, some researchers propose to use alternating direction methods of multipliers (ADMM) to directly solve (1.1) without reparametrization as a linear program. Although these methods enjoy $O(1/T)$ convergence rates based on variational inequality criteria, where $T$ is the number of iterations. ADMM can be viewed as an exterior point method, and always gives infeasible solutions within finite number of iterations. We often observe that after ADMM takes a large number of iterations, the solutions still suffer from significant feasibility violation. These methods work well only for moderate scale problems (e.g., $d < 1000$). For larger $d$, ADMM becomes less competitive.

These methods, though popular, are usually designed for solving (1.2) and (1.3) for one single regularization factor. This is not satisfactory, since an appropriate choice of $\lambda$ is usually unknown. Thus, one usually expects an algorithm to obtain multiple solutions tuned over a reasonable range of values for $\lambda$. For each value of $\lambda$, we need to solve a linear program from scratch, and it is therefore often very inefficient for high dimensional problems.

To overcome the above drawbacks, we propose to solve both (1.2) and (1.3) by a variant of the parametric simplex method (PSM) in a principled manner (Murty, 1983; Vanderbei, 1995). Specifically, the *parametric simplex method* parametrizes (1.2) and (1.3) using the unknown regularization factor as a "parameter". This eventually yields a piecewise linear solution path for a sequence of regularization factors. Such a varying parameter scheme is also called homotopy optimization in existing literature. PSM relies some special rules to iteratively choose the pair of variables to swap, which algebraically calculates the solution path during each pivoting. PSM terminates at a value of parameter, where we have successfully solved the full solution path to the original problem. Although in the worst-case scenario, PSM can take an exponential number of pivots to find an optimal solution path. Our empirical results suggest that the number of iterations is roughly linear in the number of nonzero variables for large regularization factors with sparse optima. This means that the desired sparse solutions can often be found using very few pivots.

Several optimization methods for solving (1.1) are closely related to PSM. However, there is a lack of generic design in these methods. Their methods, for example, the simplex method proposed in Yao and Lee (2014) can be viewed as a special example of our proposed PSM, where the perturbation is only considered on the right-hand-side of the inequalities in the constraints. DASSO algorithm computes the entire coefficient path of Dantzig selector by a simplex-like algorithm. Zhu et al. (2004) propose a similar algorithm which takes advantage of the piece-wise linearity of the problem and computes the whole solution path on $\ell_1$-SVM. These methods can be considered as similar algorithms derived from PSM but only applied to special cases, where the entire solution path is computed but an accurate dual certificate stopping criterion is not provided.

**Notations:** We denote all zero and all one vectors by **1** and **0** respectively. Given a vector $a = (a_1, ..., a_d)^\top \in \mathbb{R}^d$, we define the number of nonzero entries $\|a\|_0 = \sum_j 1(a_j \neq 0)$, $\|a\|_1 = \sum_j |a_j|$, $\|a\|_2^2 = \sum_j a_j^2$, and $\|a\|_\infty = \max_j |a_j|$. When comparing vectors, "$\geq$" and "$\leq$" mean component-wise comparison. Given a matrix $A \in \mathbb{R}^{d \times d}$ with entries $a_{jk}$, we use $\|A\|$ to denote entry-wise norms and $\||A\||$ to denote matrix norms. Accordingly $\|A\|_0 = \sum_{j,k} 1(a_{jk} \neq 0)$, $\|A\|_1 = \sum_{j,k} |a_{jk}|$, $\|A\|_\infty = \max_{j,k} |a_{jk}|$, $\||A\||_1 = \max_k \sum_j |a_{jk}|$, $\||A\||_\infty = \max_j \sum_k |a_{jk}|$, $\||A\||_2 = \max_{\|a\|_2 \leq 1} \|Aa\|_2$, and $\|A\|_F^2 = \sum_{j,k} a_{jk}^2$. We denote $A_{\backslash i \backslash j}$ as the submatrix of $A$ with $i$-th row and $j$-th column removed. We denote $A_{i \backslash j}$ as the $i$-th row of $A$ with its $j$-th entry removed and $A_{\backslash ij}$ as the $j$-th column of $A$ with its $i$-th entry removed. For any subset $\mathcal{G}$ of $\{1, 2, \ldots, d\}$, we let $A_{\mathcal{G}}$ denote the submatrix of $A \in \mathbb{R}^{p \times d}$ consisting of the corresponding columns of $A$. The notation $A \geq \mathbf{0}$ means all of $A$'s entries are nonnegative. Similarly, for a vector $a \in \mathbb{R}^d$, we let $a_{\mathcal{G}}$ denote the subvector of $a$ associated with the indices in $\mathcal{G}$. Finally, $I_d$ denotes the $d$-dimensional identity matrix

and $e_i$ denotes vector that has a one in its $i$-th entry and zero elsewhere. In a large matrix, we leave a submatrix blank when all of its entries are zeros.

## 2  Background

Many sparse learning approaches are formulated as convex programs in a generic form:

$$\min_\theta \mathcal{L}(\theta) + \lambda\|\theta\|_1, \tag{2.1}$$

where $\mathcal{L}(\theta)$ is a convex loss function, and $\lambda > 0$ is a regularization factor controlling bias and variance. Moreover, if $\mathcal{L}(\theta)$ is smooth, we can also consider an alternative formulation:

$$\min_\theta \|\theta\|_1 \quad \text{s.t.} \quad \|\nabla\mathcal{L}(\theta)\|_\infty \leq \lambda, \tag{2.2}$$

where $\nabla\mathcal{L}(\theta)$ is the gradient of $\mathcal{L}(\theta)$, and $\lambda > 0$ is a regularization factor. As will be shown later, both (2.2) and (2.1) are naturally suited for our algorithm, when $\mathcal{L}(\theta)$ is piecewise linear or quadratic respectively. Our algorithm yields a piecewise-linear solution path as a function of $\lambda$ by varying $\lambda$ from large to small.

Before we proceed with our proposed algorithm, we first introduce the sparse learning problems of our interests, including sparse linear regression, sparse linear classification, and undirected graph estimation. Due to space limit, we only present three examples, and defer the others to the appendix.

**Dantzig Selector:**   The first problem is sparse linear regression. Let $y \in \mathbb{R}^n$ be a response vector and $X \in \mathbb{R}^{n \times d}$ be the design matrix. We consider a linear model $y = X\theta^* + \epsilon$, where $\theta^* \in \mathbb{R}^d$ is the unknown regression coefficient vector, and $\epsilon$ is the observational noise vector. Here we are interested in a high dimensional regime: $d$ is much larger than $n$, i.e., $d \gg n$, and many entries in $\theta^*$ are zero, i.e., $\|\theta^*\|_0 = s^* \ll n$. To get a sparse estimator of $\theta^0$, machine learning researchers and statisticians have proposed numerous approaches including Lasso (Tibshirani, 1996), Dantzig Selector (Candes and Tao, 2007) and LAD-Lasso (Wang et al., 2007).

The *Dantzig selector* is formulated as the solution to the following convex program:

$$\min_\theta \|\theta\|_1 \quad \text{subject to} \quad \|X^\top(y - X\theta)\|_\infty \leq \lambda. \tag{2.3}$$

By setting $\theta = \theta^+ - \theta^-$ with $\theta_j^+ = \theta_j \cdot \mathbb{1}(\theta_j > 0)$ and $\theta_j^+ = \theta_j \cdot \mathbb{1}(\theta_j < 0)$, we rewrite (2.3) as a linear program:

$$\min_{\theta^+, \theta^-} \mathbf{1}^\top(\theta^+ + \theta^-) \quad \text{s.t.} \begin{pmatrix} X^\top X & -X^\top X \\ -X^\top X & X^\top X \end{pmatrix} \begin{pmatrix} \theta^+ \\ \theta^- \end{pmatrix} \leq \begin{pmatrix} \lambda\mathbf{1} + X^\top y \\ \lambda\mathbf{1} - X^\top y \end{pmatrix}, \quad \theta^+, \theta^- \geq \mathbf{0}. \tag{2.4}$$

By complementary slackness, we can guarantee that the optimal $\theta_j^+$'s and $\theta_j^-$'s are nonnegative and complementary to each other. Note that (2.4) fits into our parametric linear program as (1.3) with

$$A = \begin{pmatrix} X^\top X & -X^\top X \\ -X^\top X & X^\top X \end{pmatrix}, \quad b = \begin{pmatrix} X^\top y \\ -X^\top y \end{pmatrix}, \quad c = -\mathbf{1}, \quad \bar{b} = \mathbf{1}, \quad \bar{c} = \mathbf{0}, \quad x = \begin{pmatrix} \theta^+ \\ \theta^- \end{pmatrix}.$$

**Sparse Support Vector Machine:**   The second problem is Sparse SVM (Support Vector Machine), which is a model-free discriminative modeling approach (Zhu et al., 2004). Given $n$ independent and identically distributed samples $(x_1, y_1), ..., (x_n, y_n)$, where $x_i \in \mathbb{R}^d$ is the feature vector and $y_i \in \{1, -1\}$ is the binary label. Similar to sparse linear regression, we are interested in the high dimensional regime. To obtain a sparse SVM classifier, we solve the following convex program:

$$\min_{\theta_0, \theta} \sum_{i=1}^n [1 - y_i(\theta_0 + \theta^\top x_i)]_+ \quad \text{s.t.} \|\theta\|_1 \leq \lambda, \tag{2.5}$$

where $\theta_0 \in \mathbb{R}$ and $\theta \in \mathbb{R}^d$. Given a new sample $z \in \mathbb{R}^d$, Sparse SVM classifier predicts its label by $\text{sign}(\theta_0 + \theta^\top z)$. Let $t_i = 1 - y_i(\theta_0 + \theta^\top x_i)$ for $i = 1, ..., n$. Then $t_i$ can be expressed as $t_i = t_i^+ - t_i^-$. Notice $[1 - y_i(\theta_0 + \theta^\top x_i)]_+$ can be represented by $t_i^+$. We split $\theta$ and $\theta_0$ into positive and negative parts as well: $\theta = \theta^+ - \theta^-$ and $\theta_0 = \theta_0^+ + \theta_0^-$ and add slack variable $w$ to the constraint so that the constraint becomes equality: $\theta^+ + \theta^- + w = \lambda\mathbf{1}$, $w \geq \mathbf{0}$. Now we cast the problem into the equality parametric simplex form (1.2). We identify each component of (1.2) as the following: $x = \begin{pmatrix} t^+ & t^- & \theta^+ & \theta^- & \theta_0^+ & \theta_0^- & w \end{pmatrix}^\top \in \mathbb{R}^{(n+1) \times (2n+3d+2)}, x \geq \mathbf{0}$, $c = \begin{pmatrix} -\mathbf{1}^\top & \mathbf{0}^\top & \mathbf{0}^\top & \mathbf{0}^\top & 0 & 0 & \mathbf{0}^\top \end{pmatrix}^\top \in \mathbb{R}^{2n+3d+2}$, $\bar{c} = \mathbf{0} \in \mathbb{R}^{2n+3d+2}$, $b = \begin{pmatrix} \mathbf{1}^\top & 0 \end{pmatrix}^\top \in \mathbb{R}^{n+1}$, $\bar{b} = \begin{pmatrix} \mathbf{0}^\top & 1 \end{pmatrix}^\top \in \mathbb{R}^{n+1}$, and $A = \begin{pmatrix} I_n & -I_n & Z & -Z & y & -y & \\ & & \mathbf{1}^\top & \mathbf{1}^\top & & & \mathbf{1}^\top \end{pmatrix} \in \mathbb{R}^{(n+1) \times (2n+3d+2)}$, where $Z = (y_1 x_1, \ldots, y_n x_n)^\top \in \mathbb{R}^{n \times d}$.

**Differential Graph Estimation:** The third problem is the differential graph estimation, which aims to identify the difference between two undirected graphs (Zhao et al., 2013; Danaher et al., 2013). The related applications in biological and medical research can be found in existing literature (Hudson et al., 2009; Bandyopadhyaya et al., 2010; Ideker and Krogan, 2012). Specifically, given $n_1$ i.i.d. samples $x_1, ..., x_n$ from $N_d(\mu_X^0, \Sigma_X^0)$ and $n_2$ i.i.d. samples $y_1, ..., y_n$ from $N_d(\mu_Y^0, \Sigma_Y^0)$ We are interested in estimating the difference of the precision matrices of two distributions:

$$\Delta^0 = (\Sigma_X^0)^{-1} - (\Sigma_Y^0)^{-1}.$$

We define the empirical covariance matrices as: $S_X = \frac{1}{n_1} \sum_{j=1}^{n_1} (x_j - \bar{x})(x_j - \bar{x})^\top$ and $S_Y = \frac{1}{n_2} \sum_{j=1}^{n_2} (y_j - \bar{y})(y_j - \bar{x})^\top$, where $\bar{x} = \frac{1}{n_1} \sum_{j=1}^{n} x_j$ and $\bar{y} = \frac{1}{n_2} \sum_{j=1}^{n} y_j$. Then Zhao et al. (2013) propose to estimate $\Delta^0$ by solving the following problem:

$$\min_{\Delta} \|\Delta\|_1 \quad \text{s.t.} \|S_X \Delta S_Y - S_X + S_Y\|_\infty \le \lambda, \tag{2.6}$$

where $S_X$ and $S_Y$ are the empirical covariance matrices. As can be seen, (2.6) is essentially a special example of a more general parametric linear program as follows,

$$\min_{D} \|D\|_1 \quad \text{s.t.} \|XDZ - Y\|_\infty \le \lambda, \tag{2.7}$$

where $D \in \mathbb{R}^{d_1 \times d_2}$, $X \in \mathbb{R}^{m_1 \times d_1}$, $Z \in \mathbb{R}^{d_2 \times m_2}$ and $Y \in \mathbb{R}^{m_1 \times m_2}$ are given data matrices.

Instead of directly solving (2.7), we consider a reparametrization by introducing an axillary variable $C = XD$. Similar to CLIME, we decompose $D = D^+ + D^-$, and eventually rewrite (2.7) as

$$\min_{D^+, D^-} \mathbf{1}^\top (D^+ + D^-)\mathbf{1} \quad \text{s.t.} \|CZ - Y\|_\infty \le \lambda, \quad X(D^+ - D^-) = C, \quad D^+, D^- \ge \mathbf{0}, \tag{2.8}$$

Let $\text{vec}(D^+)$, $\text{vec}(D^-)$, $\text{vec}(C)$ and $\text{vec}(Y)$ be the vectors obtained by stacking the columns of matrices $D^+, D^-\ C$ and $Y$, respectively. We write (2.8) as a parametric linear program,

$$A = \begin{pmatrix} X^0 & -X^0 & -I_{m_1 d_2} & \\ & & Z^0 & I_{m_1 m_2} \\ & & -Z^0 & & I_{m_1 m_2} \end{pmatrix}$$

with $X^0 = \begin{pmatrix} X & & \\ & \ddots & \\ & & X \end{pmatrix} \in \mathbb{R}^{m_1 d_2 \times d_1 d_2}$, $Z^0 = \begin{pmatrix} z_{11} I_{m_1} & \cdots & z_{d_2 1} I_{m_1} \\ \vdots & \ddots & \vdots \\ -z_{1 m_2} I_{m_1} & \cdots & -z_{d_2 m_2} I_{m_1} \end{pmatrix} \in$

$\mathbb{R}^{m_1 m_2 \times m_1 d_2}$, where $z_{ij}$ denotes the $(i, j)$ entry of matrix $Z$;

$$x = \begin{bmatrix} \text{vec}(D^+) & \text{vec}(D^-) & \text{vec}(C) & w \end{bmatrix}^\top \in \mathbb{R}^{2d_1 d_2 + m_1 d_2 + 2m_1 m_2},$$

where $w \in \mathbb{R}^{2m_1 m_2}$ is nonnegative slack variable vector used to make the inequality become an equality. Moreover, we also have $b = \begin{bmatrix} \mathbf{0}^\top & \text{vec}(Y) & -\text{vec}(Y) \end{bmatrix}^\top \in \mathbb{R}^{m_1 d_2 + 2m_1 m_2}$, where the first $m_1 d_2$ components of vector $b$ are $0$ and the rest components are from matrix $Y$; $\bar{b} = \begin{pmatrix} \mathbf{0}^\top & \mathbf{1}^\top & \mathbf{1}^\top \end{pmatrix}^\top \in \mathbb{R}^{m_1 d_2 + 2m_1 m_2}$, where the first $m_1 d_2$ components of vector $\bar{b}$ are $0$ and the rest $2m_1 m_2$ components are 1; $c = \begin{pmatrix} -\mathbf{1}^\top & -\mathbf{1}^\top & \mathbf{0}^\top & \mathbf{0}^\top \end{pmatrix}^\top \in \mathbb{R}^{2d_1 d_2 + m_1 d_2 + 2m_1 m_2}$, where the first $2d_1 d_2$ components of vector $c$ are $-1$ and the rest $m_1 d_2 + 2m_1 m_2$ components are 0.

# 3 Homotopy Parametric Simplex Method

We first briefly review the primal simplex method for linear programming, and then derive the proposed algorithm.

**Preliminaries:** We consider a standard linear program as follows,

$$\max_{x} c^\top x \quad \text{s.t.} Ax = b, \quad x \ge \mathbf{0} \quad x \in \mathbb{R}^n, \tag{3.1}$$

where $A \in \mathbb{R}^{m \times n}$, $b \in \mathbb{R}^m$ and $c \in \mathbb{R}^n$ are given. Without loss of generality, we assume that $m \le n$ and matrix $A$ has full row rank $m$. Throughout our analysis, we assume that an optimal solution exists (it needs not be unique). The *primal simplex method* starts from a basic feasible solution (to be defined shortly—but geometrically can be thought of as any vertex of the feasible polytope) and proceeds step-by-step (vertex-by-vertex) to the optimal solution. Various techniques exist to find the first feasible solution, which is often referred to the Phase I method. See Vanderbei (1995); Murty (1983); Dantzig (1951).

Algebraically, a *basic solution* corresponds to a partition of the indices $\{1, \ldots, n\}$ into $m$ *basic indices* denoted $\mathcal{B}$ and $n - m$ *non-basic indices* denoted $\mathcal{N}$. Note that not all partitions are allowed—the submatrix of $A$ consisting of the columns of $A$ associated with the basic indices, denoted $A_{\mathcal{B}}$, must be invertible. The submatrix of $A$ corresponding to the nonbasic indices is denoted $A_{\mathcal{N}}$. Suppressing the fact that the columns have been rearranged, we can write $A = [A_{\mathcal{N}}, \ A_{\mathcal{B}}]$. If we rearrange the rows of $x$ and $c$ in the same way, we can introduce a corresponding partition of these vectors: $x = \begin{bmatrix} x_{\mathcal{N}} \\ x_{\mathcal{B}} \end{bmatrix}$, $c = \begin{bmatrix} c_{\mathcal{N}} \\ c_{\mathcal{B}} \end{bmatrix}$. From the commutative property of addition, we rewrite the constraint as $A_{\mathcal{N}} x_{\mathcal{N}} + A_{\mathcal{B}} x_{\mathcal{B}} = b$. Since the matrix $A_{\mathcal{B}}$ is assumed to be invertible, we can express $x_{\mathcal{B}}$ in terms of $x_{\mathcal{N}}$ as follows:

$$x_{\mathcal{B}} = x_{\mathcal{B}}^* - A_{\mathcal{B}}^{-1} A_{\mathcal{N}} x_{\mathcal{N}}, \tag{3.2}$$

where we have written $x_{\mathcal{B}}^*$ as an abbreviation for $A_{\mathcal{B}}^{-1} b$. This rearrangement of the equality constraints is called a *dictionary* because the basic variables are defined as functions of the nonbasic variables.

Denoting the objective $c^\top x$ by $\zeta$, then we also can write:

$$\zeta = c^\top x = c_{\mathcal{B}}^\top x_{\mathcal{B}} + c_{\mathcal{N}}^\top x_{\mathcal{N}} = \zeta^* - (z_{\mathcal{N}}^*)^\top x_{\mathcal{N}}, \tag{3.3}$$

where $\zeta^* = c_{\mathcal{B}}^\top A_{\mathcal{B}}^{-1} b$, $x_{\mathcal{B}}^* = A_{\mathcal{B}}^{-1} b$ and $z_{\mathcal{N}}^* = (A_{\mathcal{B}}^{-1} A_{\mathcal{N}})^\top c_{\mathcal{B}} - c_{\mathcal{N}}$.

We call equations (3.2) and (3.3) the *primal dictionary* associated with the current basis $\mathcal{B}$. Corresponding to each dictionary, there is a *basic solution* (also called a dictionary solution) obtained by setting the nonbasic variables to zero and reading off values of the basic variables: $x_{\mathcal{N}} = \mathbf{0}$, $\quad x_{\mathcal{B}} = x_{\mathcal{B}}^*$. This particular "solution" satisfies the equality constraints of the problem by construction. To be a feasible solution, one only needs to check that the values of the basic variables are nonnegative. Therefore, we say that a basic solution is a *basic feasible solution* if $x_{\mathcal{B}}^* \geq \mathbf{0}$.

The dual of (3.1) is given by

$$\max_y -b^\top y \quad \text{s.t. } A^\top y - z = c, \quad z \geq \mathbf{0} \quad z \in \mathbb{R}^n, y \in \mathbb{R}^m. \tag{3.4}$$

In this case, we separate variable $z$ into basic and nonbasic parts as before: $[z] = \begin{bmatrix} z_{\mathcal{N}} \\ z_{\mathcal{B}} \end{bmatrix}$. The corresponding *dual dictionary* is given by:

$$z_{\mathcal{N}} = z_{\mathcal{N}}^* + (A_{\mathcal{B}}^{-1} A_{\mathcal{N}})^\top z_{\mathcal{B}}, \quad -\xi = -\zeta^* - (x_{\mathcal{B}}^*)^\top z_{\mathcal{B}}, \tag{3.5}$$

where $\xi$ denotes the objective function in the (3.4), $\zeta^* = c_{\mathcal{B}}^\top A_{\mathcal{B}}^{-1} b$, $x_{\mathcal{B}}^* = A_{\mathcal{B}}^{-1} b$ and $z_{\mathcal{N}}^* = (A_{\mathcal{B}}^{-1} A_{\mathcal{N}})^\top c_{\mathcal{B}} - c_{\mathcal{N}}$.

For each dictionary, we set $x_{\mathcal{N}}$ and $z_{\mathcal{B}}$ to $\mathbf{0}$ (complementarity) and read off the solutions to $x_{\mathcal{B}}$ and $z_{\mathcal{N}}$ according to (3.2) and (3.5). Next, we remove one basic index and replacing it with a nonbasic index, and then get an updated dictionary. The simplex method produces a sequence of steps to *adjacent* bases such that the value of the objective function is always increasing at each step. Primal feasibility requires that $x_{\mathcal{B}} \geq \mathbf{0}$, so while we update the dictionary, primal feasibility must always be satisfied. This process will stop when $z_{\mathcal{N}} \geq \mathbf{0}$ (dual feasibility), since it satisfies primal feasibility, dual feasibility and complementarity (i.e., the optimality condition).

**Parametric Simplex Method:** We derive the parametric simplex method used to find the full solution path while solving the parametric linear programming problem only once. A few variants of the simplex method are proposed with different rules for choosing the pair of variables to swap at each iteration. Here we describe the rule used by the parametric simplex method: we add some positive perturbations ($\bar{b}$ and $\bar{c}$) times a positive parameter $\lambda$ to both objective function and the right hand side of the primal problem. The purpose of doing this is to guarantee the primal and dual feasibility when $\lambda$ is large. Since the problem is already primal feasible and dual feasible, there is no phase I stage required for the parametric simplex method. Furthermore, if the $i$-th entry of $b$ or the $j$-th entry of $c$ has already satisfied the feasibility condition ($b_i \geq \mathbf{0}$ or $c_j \leq \mathbf{0}$), then the corresponding perturbation $\bar{b}_i$ or $\bar{c}_j$ to that entry is allowed to be $0$. With these perturbations, (3.1) becomes:

$$\max_x (c + \lambda \bar{c})^\top x \quad \text{s.t. } Ax = b + \lambda \bar{b}, \quad x \geq \mathbf{0} \quad x \in \mathbb{R}^n. \tag{3.6}$$

We separate the perturbation vectors into basic and nonbasic parts as well and write down the the dictionary with perturbations corresponding to (3.2),(3.3), and (3.5) as:

$$x_{\mathcal{B}} = (x_{\mathcal{B}}^* + \lambda \bar{x}_{\mathcal{B}}) - A_{\mathcal{B}}^{-1} A_{\mathcal{N}} x_{\mathcal{N}}, \quad \zeta = \zeta^* - (z_{\mathcal{N}}^* + \lambda \bar{z}_{\mathcal{N}})^\top x_{\mathcal{N}}, \tag{3.7}$$

$$z_{\mathcal{N}} = (z_{\mathcal{N}}^* + \lambda \bar{z}_{\mathcal{N}}) + (A_{\mathcal{B}}^{-1} A_{\mathcal{N}})^\top z_{\mathcal{B}}, \quad -\xi = -\zeta^* - (x_{\mathcal{B}}^* + \lambda \bar{x}_{\mathcal{B}})^\top z_{\mathcal{B}}, \tag{3.8}$$

where $x_{\mathcal{B}}^* = A_{\mathcal{B}}^{-1} b$, $z_{\mathcal{N}}^* = (A_{\mathcal{B}}^{-1} A_{\mathcal{N}})^\top c_{\mathcal{B}} - c_{\mathcal{N}}$, $\bar{x}_{\mathcal{B}} = A_{\mathcal{B}}^{-1} \bar{b}$ and $\bar{z}_{\mathcal{N}} = (A_{\mathcal{B}}^{-1} A_{\mathcal{N}})^\top \bar{c}_{\mathcal{B}} - \bar{c}_{\mathcal{N}}$.

When $\lambda$ is large, the dictionary will be both primal and dual feasible ($x_{\mathcal{B}}^* + \lambda \bar{x}_{\mathcal{B}} \geq \mathbf{0}$ and $z_{\mathcal{N}}^* + \lambda \bar{z}_{\mathcal{N}} \geq \mathbf{0}$). The corresponding primal solution is simple: $x_{\mathcal{B}} = x_{\mathcal{B}}^* + \lambda \bar{x}_{\mathcal{B}}$ and $x_{\mathcal{N}} = \mathbf{0}$. This solution is valid until $\lambda$ hits a lower bound which breaks the feasibility. The smallest value of $\lambda$ without break any feasibility is given by

$$\lambda^* = \min\{\lambda : z_{\mathcal{N}}^* + \lambda \bar{z}_{\mathcal{N}} \geq \mathbf{0} \text{ and } x_{\mathcal{B}}^* + \lambda \bar{x}_{\mathcal{B}} \geq \mathbf{0}\}. \tag{3.9}$$

In other words, the dictionary and its corresponding solution $x_{\mathcal{B}} = x_{\mathcal{B}}^* + \lambda \bar{x}_{\mathcal{B}}$ and $x_{\mathcal{N}} = \mathbf{0}$ is optimal for the value of $\lambda \in [\lambda^*, \lambda_{\max}]$, where

$$\lambda^* = \max \quad \left( \max_{j \in \mathcal{N}, \, \bar{z}_{\mathcal{N}_j} > 0} -\frac{z_{\mathcal{N}_j}^*}{\bar{z}_{\mathcal{N}_j}}, \quad \max_{i \in \mathcal{B}, \bar{x}_{\mathcal{B}_i} > 0} -\frac{x_{\mathcal{B}_i}^*}{\bar{x}_{\mathcal{B}_i}} \right), \tag{3.10}$$

$$\lambda_{\max} = \min \quad \left( \min_{j \in \mathcal{N}, \, \bar{z}_{\mathcal{N}_j} < 0} -\frac{z_{\mathcal{N}_j}^*}{\bar{z}_{\mathcal{N}_j}}, \quad \min_{i \in \mathcal{B}, \bar{x}_{\mathcal{B}_i} < 0} -\frac{x_{\mathcal{B}_i}^*}{\bar{x}_{\mathcal{B}_i}} \right). \tag{3.11}$$

Note that although initially the perturbations are nonnegative, as the dictionary gets updated, the perturbation does not necessarily maintain nonnegativity. For each dictionary, there is a corresponding interval of $\lambda$ given by (3.10) and (3.11). We have characterized the optimal solution for this interval, and these together give us the solution path of the original parametric linear programming problem. Next, we show how the dictionary gets updated as the leaving variable and entering variable swap.

We expect that after swapping the entering variable $j$ and leaving variable $i$, the new solution in the dictionary (3.7) and (3.8) would slightly change to:

$$x_j^* = t, \quad \bar{x}_j^* = \bar{t}, \quad z_i^* = s, \quad \bar{z}_i^* = \bar{s}, \quad x_{\mathcal{B}}^* \leftarrow x_{\mathcal{B}}^* - t\Delta x_{\mathcal{B}}, \quad \bar{x}_{\mathcal{B}} \leftarrow \bar{x}_{\mathcal{B}} - \bar{t}\Delta x_{\mathcal{B}},$$

$$z_{\mathcal{N}}^* \leftarrow z_{\mathcal{N}}^* - s\Delta z_{\mathcal{N}}, \quad \bar{z}_{\mathcal{N}} \leftarrow \bar{z}_{\mathcal{N}} - \bar{s}\Delta z_{\mathcal{N}},$$

where $t$ and $\bar{t}$ are the primal step length for the primal basic variables and perturbations, $s$ and $\bar{s}$ are the dual step length for the dual nonbasic variables and perturbations, $\Delta x_{\mathcal{B}}$ and $\Delta z_{\mathcal{N}}$ are the primal and dual step directions, respectively. We explain how to find these values in details now.

There is either a $j \in \mathcal{N}$ for which $z_{\mathcal{N}}^* + \lambda \bar{z}_{\mathcal{N}} = 0$ or an $i \in \mathcal{B}$ for which $x_{\mathcal{B}}^* + \lambda \bar{x}_{\mathcal{B}} = 0$ in (3.9). If it corresponds to a nonbasic index $j$, then we do one step of the primal simplex. In this case, we declare $j$ as the entering variable, then we need to find the primal step direction $\Delta x_{\mathcal{B}}$. After the entering variable $j$ has been selected, $x_{\mathcal{N}}$ changes from $\mathbf{0}$ to $te_j$, where $t$ is the primal step length. Then according to (3.7), we have that $x_{\mathcal{B}} = (x_{\mathcal{B}}^* + \lambda \bar{x}_{\mathcal{B}}) - A_{\mathcal{B}}^{-1} A_{\mathcal{N}} te_j$. The step direction $\Delta x_{\mathcal{B}}$ is given by $\Delta x_{\mathcal{B}} = A_{\mathcal{B}}^{-1} A_{\mathcal{N}} e_j$. We next select the leaving variable. In order to maintain primal feasibility, we need to keep $x_{\mathcal{B}} \geq \mathbf{0}$, therefore, the leaving variable $i$ is selected such that $i \in \mathcal{B}$ achieves the maximal value of $\frac{\Delta x_i}{x_i^* + \lambda^* \bar{x}_i}$. It only remains to show how $z_{\mathcal{N}}$ changes. Since $i$ is the leaving variable, according to (3.8), we have $\Delta z_{\mathcal{N}} = -(A_{\mathcal{B}}^{-1} A_{\mathcal{N}})^\top e_i$. After we know the entering variables, the primal and dual step directions, the primal and dual step lengths can be found as $t = \frac{x_i^*}{\Delta x_i}, \quad \bar{t} = \frac{\bar{x}_i}{\Delta x_i}, \quad s = \frac{z_j^*}{\Delta z_j}, \quad \bar{s} = \frac{\bar{z}_j}{\Delta z_j}$.

If, on the other hand, the constraint in (3.9) corresponds to a basic index $i$, we declare $i$ as the leaving variable, then similar calculation can be made based on the dual simplex method (apply the primal simplex method to the dual problem). Since it is very similar to the primal simplex method, we omit the detailed description.

The algorithm will terminate whenever $\lambda^* \leq 0$. The corresponding solution is optimal since our dictionary always satisfies primal feasibility, dual feasibility and complementary slackness condition. The only concern during the entire process of the parametric simplex method is that $\lambda$ does not equal to zero, so as long as $\lambda$ can be set to be zero, we have the optimal solution to the original problem. We summarize the parametric simplex method in Algorithm 1:

The following theorem shows that the updated basic and nonbasic partition gives the optimal solution.

**Theorem 3.1.** For a given dictionary with parameter $\lambda$ in the form of (3.7) and (3.8), let $\mathcal{B}$ be a basic index set and $\mathcal{N}$ be an nonbasic index set. Assume this dictionary is optimal for $\lambda \in [\lambda^*, \lambda_{\max}]$, where $\lambda^*$ and $\lambda_{\max}$ are given by (3.10) and (3.11), respectively. The updated dictionary with basic index set $\mathcal{B}^*$ and nonbasic index set $\mathcal{N}^*$ is still optimal at $\lambda = \lambda^*$.

Write down the dictionary as in (3.7) and (3.8);
Find $\lambda^*$ given by (3.10);
**while** $\lambda^* > 0$ **do**

$\quad$ **if** *the constraint in* (3.10) *corresponds to an index* $j \in \mathcal{N}$ **then**

$\qquad$ Declare $x_j$ as the entering variable;

$\qquad$ Compute primal step direction. $\Delta x_{\mathcal{B}} = A_{\mathcal{B}}^{-1} A_{\mathcal{N}} e_j$;

$\qquad$ Select leaving variable. Need to find $i \in \mathcal{B}$ that achieves the maximal value of $\frac{\Delta x_i}{x_i^* + \lambda^* \bar{x}_i}$;

$\qquad$ Compute dual step direction. It is given by $\Delta z_{\mathcal{N}} = -(A_{\mathcal{B}}^{-1} A_{\mathcal{N}})^\top e_i$;

$\quad$ **else if** *the constraint in* (3.10) *corresponds to an index* $i \in \mathcal{B}$ **then**

$\qquad$ Declare $z_i$ as the leaving variable;

$\qquad$ Compute dual step direction. $\Delta z_{\mathcal{N}} = -(A_{\mathcal{B}}^{-1} A_{\mathcal{N}})^\top e_i$;

$\qquad$ Select entering variable. Need to find $j \in \mathcal{N}$ that achieves the maximal value of $\frac{\Delta z_j}{z_j^* + \lambda^* \bar{z}_j}$;

$\qquad$ Compute primal step direction. It is given by $\Delta x_{\mathcal{B}} = A_{\mathcal{B}}^{-1} A_{\mathcal{N}} e_j$;

$\quad$ Compute the dual and primal step lengths for both variables and perturbations:

$$t = \frac{x_i^*}{\Delta x_i}, \quad \bar{t} = \frac{\bar{x}_i}{\Delta x_i}, \quad s = \frac{z_j^*}{\Delta z_j}, \quad \bar{s} = \frac{\bar{z}_j}{\Delta z_j}.$$

$\quad$ Update the primal and dual solutions:

$$x_j^* = t, \quad \bar{x}_j = \bar{t}, \quad z_i^* = s, \quad \bar{z}_i = \bar{s},$$

$$x_{\mathcal{B}}^* \leftarrow x_{\mathcal{B}}^* - t\Delta x_{\mathcal{B}}, \quad \bar{x}_{\mathcal{B}} \leftarrow \bar{x}_{\mathcal{B}} - \bar{t}\Delta x_{\mathcal{B}} \quad z_{\mathcal{N}}^* \leftarrow z_{\mathcal{N}}^* - s\Delta z_{\mathcal{N}}, \quad \bar{z}_{\mathcal{N}} \leftarrow \bar{z}_{\mathcal{N}} - \bar{s}\Delta z_{\mathcal{N}}.$$

$\quad$ Update the basic and nonbasic index sets $\mathcal{B} := \mathcal{B} \setminus \{i\} \cap \{j\}$ and $\mathcal{N} := \mathcal{N} \setminus \{j\} \cap \{i\}$. Write down the new dictionary and compute $\lambda^*$ given by (3.10);

**end**
Set the nonbasic variables as **0**s and read the values of the basic variables.

**Algorithm 1:** The parametric simplex method

During each iteration, there is an optimal solution corresponding to $\lambda \in [\lambda^*, \lambda_{\max}]$. Notice each of these $\lambda$'s range is determined by a partition between basic and nonbasic variables, and the number of the partition into basic and nonbasic variables is finite. Thus after finite steps, we must find the optimal solution corresponding to all $\lambda$ values.

**Theory:** We present our theoretical analysis on solving Dantzig selector using PSM. Specifically, given $X \in \mathbb{R}^{n \times d}$, $y \in \mathbb{R}^n$, we consider a linear model $y = X\theta^* + \epsilon$, where $\theta^*$ is the unknown sparse regression coefficient vector with $\|\theta^*\|_0 = s^*$, and $\epsilon \sim N(0, \sigma^2 I_n)$. We show that PSM always maintains a pair of sparse primal and dual solutions. Therefore, the computation cost within each iteration of PSM can be significantly reduced. Before we proceed with our main result, we introduce two assumptions. The first assumption requires the regularization factor to be sufficiently large.

**Assumption 3.2.** Suppose that PSM solves (2.3) for a regularization sequence $\{\lambda_K\}_{K=0}^N$. The smallest regularization factor $\lambda_N$ satisfies

$$\lambda_N = C\sigma\sqrt{\frac{\log d}{n}} \geq 4\|X^\top \epsilon\|_\infty \quad \text{for some generic constant } C.$$

Existing literature has extensively studied Assumption 3.2 for high dimensional statistical theories. Such an assumption enforces all regularization parameters to be sufficiently large in order to eliminate irrelevant coordinates along the regularization path. Note that Assumption 3.2 is deterministic for any given $\lambda_N$. Existing literature has verified that for sparse linear regression models, given $\epsilon \sim N(0, \sigma^2 I_n)$, Assumption 3.2 holds with overwhelming probability.

Before we present the second assumption, we define the largest and smallest $s$-sparse eigenvalues of $n^{-1} X^\top X$ respectively as follows.

**Definition 3.3.** Given an integer $s \geq 1$, we define

$$\rho_+(s) = \sup_{\|\Delta\|_0 \leq s} \frac{\Delta^T X^\top X \Delta}{n\|\Delta\|_2^2} \quad \text{and} \quad \rho_-(s) = \inf_{\|\Delta\|_0 \leq s} \frac{\Delta^T X^\top X \Delta}{n\|\Delta\|_2^2}.$$

**Assumption 3.4.** Given $\|\theta^*\|_0 \leq s^*$, there exists an integer $\widetilde{s}$ such that
$$\widetilde{s} \geq 100\kappa s^*, \ \ \rho_+(s^* + \widetilde{s}) < +\infty, \ \ and \ \ \widetilde{\rho}_-(s^* + \widetilde{s}) > 0,$$
where $\kappa$ is defined as $\kappa = \rho_+(s^* + \widetilde{s})/\widetilde{\rho}_-(s^* + \widetilde{s})$.

Assumption 3.4 guarantees that $n^{-1}X^\top X$ satisfies the sparse eigenvalue conditions as long as the number of active irrelevant blocks never exceeds $\widetilde{2s}$ along the solution path. That is closely related to the restricted isometry property (RIP) and restricted eigenvalue (RE) conditions, which have been extensively studied in existing literature.

We then characterize the sparsity of the primal and dual solutions within each iteration.

**Theorem 3.5** (Primal and Dual Sparsity). Suppose that Assumptions 3.2 and 3.4 hold. We consider an alternative formulation to the Dantzig selector,
$$\widehat{\theta}^{\lambda'} = \underset{\theta}{\operatorname{argmin}} \|\theta\|_1 \quad \text{subject to} \ - \nabla_j \mathcal{L}(\theta) \leq \lambda', \ \nabla_j \mathcal{L}(\theta) \leq \lambda'. \tag{3.12}$$
Let $\widehat{\mu}^{\lambda'} = [\widehat{\mu}_1^{\lambda'}, ..., \widehat{\mu}_d^{\lambda'}, \widehat{\gamma}_{d+1}^{\lambda'}, ..., \widehat{\gamma}_{2d}^{\lambda'}]^\top$ denote the optimal dual variables to (3.12). For any $\lambda' \geq \lambda$, we have $\|\widehat{\mu}^{\lambda'}\|_0 + \|\widehat{\gamma}^{\lambda'}\|_0 \leq s^* + \widetilde{s}$. Moreover, given design matrix satisfying
$$\|X_{\overline{\mathcal{S}}}^\top X_{\mathcal{S}}(X_{\mathcal{S}}^\top X_{\mathcal{S}})^{-1}\|_\infty \leq 1 - \zeta,$$
where $\zeta > 0$ is a generic constant, $\mathcal{S} = \{j \mid \theta_j^* \neq 0\}$ and $\overline{\mathcal{S}} = \{j \mid \theta_j^* = 0\}$, we have $\|\widehat{\theta}^{\lambda'}\|_0 \leq s^*$.

The proof of Theorem 3.5 is provided in Appendix B. Theorem 3.5 shows that within each iteration, both primal and dual variables are sparse, i.e., the number of nonzero entries are far smaller than $d$. Therefore, the computation cost within each iteration of PSM can be significantly reduced by a factor of $O(d/s^*)$. This partially justifies the superior performance of PSM in sparse learning.

## 4 Numerical Experiments

In this section, we present some numerical experiments and give some insights about how the parametric simplex method solves different linear programming problems. We verify the following assertions: (1) The parametric simplex method requires very few iterations to identify the nonzero component if the original problem is sparse. (2) The parametric simplex method is able to find the full solution path with high precision by solving the problem only once in an efficient and scalable manner. (3) The parametric simplex method maintains the feasibility of the problem up to machine precision along the solution path.

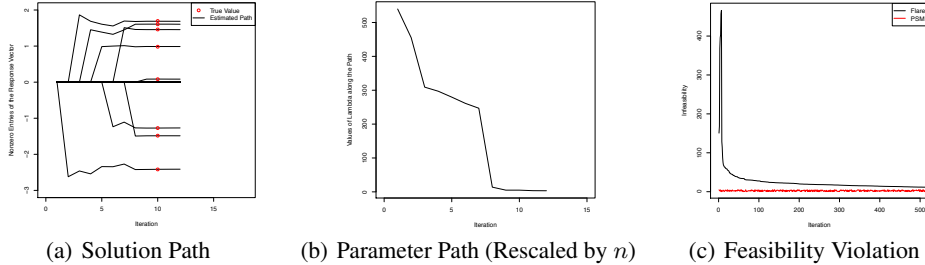

(a) Solution Path      (b) Parameter Path (Rescaled by $n$)      (c) Feasibility Violation

**Figure 1:** Dantzig selector method: (a) The solution path of the parametric simplex method; (b) The parameter path of the parametric simplex method; (c) Feasibility violation along the solution path.

**Solution path of Dantzig selector:** We start with a simple example that illustrates how the recovered solution path of the Dantzig selector model changes as the parametric simplex method iterates. We adopt the example used in Candes and Tao (2007). The design matrix $X$ has $n = 100$ rows and $d = 250$ columns. The entries of $X$ are generated from an array of independent Gaussian random variables that are then Gaussianized so that each column has a given norm. We randomly select $s = 8$ entries from the response vector $\theta^0$, and set them as $\theta_i^0 = s_i(1 + a_i)$, where $s_i = 1$ or $-1$, with probability $1/2$ and $a_i \sim \mathcal{N}(0, 1)$. The other entries of $\theta^0$ are set to zero. We form $y = X\theta^0 + \epsilon$, where $\epsilon_i \sim \mathcal{N}(0, \sigma)$, with $\sigma = 1$. We stop the parametric simplex method when $\lambda \leq \sigma n \sqrt{\log d/n}$. The solution path of the result is shown in Figure 1(a). We see that our method correctly identifies all nonzero entries of $\theta$ in less than 10 iterations. Some small overestimations occur in a few iterations after all nonzero entries have been identified. We also show how the parameter $\lambda$ evolves as the parametric simplex method iterates in Figure 1(b). As we see, $\lambda$ decreases sharply to less than 5

after all nonzero components have been identified. This reconciles with the theorem we developed. The algorithm itself only requires a very small number of iterations to correctly identify the nonzero entries of $\theta$. In our example, each iteration in the parametric simplex method identifies one or two non-sparse entries in $\theta$.

**Feasibility of Dantzig Selector:** Another advantage of the parametric simplex method is that the solution is always feasible along the path while other estimating methods usually generate infeasible solutions along the path. We compare our algorithm with "flare" (Li et al., 2015) which uses the Alternating Direction Method of Multipliers (ADMM) using the same example described above. We compute the values of $\|X^\top X\theta^i - X^\top y\|_\infty - \lambda_i$ along the solution path, where $\theta^i$ is the $i$-th basic solution (with corresponding $\lambda_i$) obtained while the parametric simplex method is iterating. Without any doubts, we always obtain 0s during each iteration. We plug the same list of $\lambda_i$ into "flare" and compute the solution path for this list as well. As shown in Table 1, the parametric simplex method is always feasible along the path since it is solving each iteration up to machine precision; while the solution path of the ADMM is almost always breaking the feasibility by a large amount, especially in the first few iterations which correspond to large $\lambda$ values. Each experiment is repeated for 100 times.

**Table 1:** Average feasibility violation with standard errors along the solution path

|       | Maximum violation | Minimum Violation |
|-------|-------------------|-------------------|
| ADMM  | 498(122)          | 143(73.2)         |
| PSM   | 0(0)              | 0(0)              |

**Performance Benchmark of Dantzig Selector:** In this part, we compare the timing performance of our algorithm with R package "flare". We fix the sample size $n$ to be 200 and vary the data dimension $d$ from 100 to 5000. Again, each entries of $X$ is independent Gaussian and Gaussianized such that the column has uniform norm. We randomly select 2% entries from vector $\theta$ to be nonzero and each entry is chosen as $\sim \mathcal{N}(0,1)$. We compute $y = X\theta + \epsilon$, with $\epsilon_i \sim \mathcal{N}(0,1)$ and try to recover vector $\theta$, given $X$ and $y$. Our method stops when $\lambda$ is less than $2\sigma\sqrt{\log d/n}$, such that the full solution path for all the values of $\lambda$ up to this value is computed by the parametric simplex method. In "flare", we estimate $\theta$ when $\lambda$ is equal to the value in the Dantzig selector model. This means "flare" has much less computation task than the parametric simplex method. As we can see in Table 2, our method has a much better performance than "flare" in terms of speed. We compare and present the timing performance of the two algorithms in seconds and each experiment is repeated for 100 times. In practice, only very few iterations is required when the response vector $\theta$ is sparse.

**Table 2:** Average timing performance (in seconds) with standard errors in the parentheses on Dantzig selector

|       | 500          | 1000        | 2000        | 5000       |
|-------|--------------|-------------|-------------|------------|
| Flare | 19.5(2.72)   | 44.4(2.54)  | 142(11.5)   | 1500(231)  |
| PSM   | 2.40(0.220)  | 29.7(1.39)  | 47.5(2.27)  | 649(89.8)  |

**Performance Benchmark of Differential Network:** We now apply this optimization method to the Differential Network model. We need the difference between two inverse covariance matrices to be sparse. We generate $\Sigma_x^0 = U^\top \Lambda U$, where $\Lambda \in \mathbb{R}^{d\times d}$ is a diagonal matrix and its entries are i.i.d. and uniform on $[1,2]$, and $U \in \mathbb{R}^{d\times d}$ is a random matrix with i.i.d. entries from $\mathcal{N}(0,1)$. Let $D_1 \in \mathbb{R}^{d\times d}$ be a random sparse symmetric matrix with a certain sparsity level. Each entry of $D_1$ is i.i.d. and from $\mathcal{N}(0,1)$. We set $D = D_1 + 2|\lambda_{\min}(D_1)|I_d$ in order to guarantee the positive definiteness of $D$, where $\lambda_{\min}(D_1)$ is the smallest eigenvalue of $D_1$. Finally, we let $\Omega_x^0 = (\Sigma_x^0)^{-1}$ and $\Omega_y^0 = \Omega_x^0 + D$.

We then generate data of sample size $n = 100$. The corresponding sample covariance matrices $S_X$ and $S_Y$ are also computed based on the data. We are not able to find other software which can efficiently solve this problem, so we only list the timing performance of our algorithm as dimension $d$ varies from 25 to 200 in Table 3. We stop our algorithm whenever the solution achieved the desired sparsity level. When $d = 25$, 50 and 100, the sparsity level of $D_1$ is set to be 0.02 and when $d = 150$ and 200, the sparsity level of $D_1$ is set to be 0.002. Each experiment is repeated for 100 times.

**Table 3:** Average timing performance (in seconds) and iteration numbers with standard errors in the parentheses on differential network

|                  | 25                | 50             | 100         | 150          | 200         |
|------------------|-------------------|----------------|-------------|--------------|-------------|
| Timing           | 0.0185(0.00689)   | 0.376(0.124)   | 6.81(2.38)  | 13.41(1.26)  | 46.88(7.24) |
| Iteration Number | 15.5(7.00)        | 55.3(18.8)     | 164(58.2)   | 85.8(16.7)   | 140(26.2)   |

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
