[Supplementary Material]

# A  Proof of Theorem 3.1

*Proof.* In order to prove that the new dictionary is still optimal, we only need to show that new dictionary is still primal and dual feasible: $x_{\mathcal{B}^*}^* + \lambda^* \bar{x}_{\mathcal{B}} \geq \mathbf{0}$ and $z_{\mathcal{N}^*}^* + \lambda^* \bar{z}_{\mathcal{N}^*} \geq \mathbf{0}$.

Case I. When calculating $\lambda^*$ given by (3.10), if the constraint corresponds to an index $i \in \mathcal{B}$, then $z_{\mathcal{N}^*}^* + \lambda^* \bar{z}_{\mathcal{N}} \geq \mathbf{0}$ is guaranteed by the way of choosing entering variable. It remains to show the primal solution is not changed: $x_{\mathcal{B}}^* + \lambda^* \bar{x}_{\mathcal{B}} = x_{\mathcal{B}^*}^* + \lambda^* \bar{x}_{\mathcal{B}^*}$.

We observe that $A_{\mathcal{B}^*}$ is obtained by changing one column of $A_{\mathcal{B}}$ to another column vector from $A_{\mathcal{N}}$, and we assume the difference of these two vectors are $u$. Without loss of generality, we assume that the $k$-th column of $A_{\mathcal{B}}$ is replaced, now we have $A_{\mathcal{B}^*} = A_{\mathcal{B}} + ue_k^\top$. Sherman-Morrison formula says that

$$A_{\mathcal{B}} A_{\mathcal{B}^*}^{-1} = I - \frac{ue_k^\top A_{\mathcal{B}}^{-1}}{1 + e_k^\top A_{\mathcal{B}}^{-1} u} = I - \theta ue_k^\top A_{\mathcal{B}}^{-1}, \tag{A.1}$$

where $\theta = \frac{1}{1+e_k^\top A_{\mathcal{B}}^{-1} u}$. Now consider the following term:

$$\begin{aligned}
&A_{\mathcal{B}}[x_{\mathcal{B}}^* - x_{\mathcal{B}^*}^* + \lambda^*(\bar{x}_{\mathcal{B}} - \bar{x}_{\mathcal{B}^*})] \\
=&A_{\mathcal{B}}(A_{\mathcal{B}}^{-1} - A_{\mathcal{B}^*}^{-1})b + \lambda^* A_{\mathcal{B}}(A_{\mathcal{B}}^{-1} - A_{\mathcal{B}^*}^{-1})\bar{b} \\
=&b - A_{\mathcal{B}} A_{\mathcal{B}^*}^{-1} b + \lambda^*(\bar{b} - A_{\mathcal{B}} A_{\mathcal{B}^*}^{-1} \bar{b}) \\
=&(\theta ue_k^\top A_{\mathcal{B}}^{-1})b + \lambda^*(\theta ue_k^\top A_{\mathcal{B}}^{-1})\bar{b} \\
=&\theta(ue_k^\top A_{\mathcal{B}}^{-1} b + \lambda^* ue_k^\top A_{\mathcal{B}}^{-1} \bar{b}).
\end{aligned} \tag{A.2}$$

Recall in this case, we have

$$\lambda^* = \max_{i \in \mathcal{B}, \bar{x}_{\mathcal{B}_i} > 0} -\frac{x_{\mathcal{B}_i}^*}{\bar{x}_{\mathcal{B}_i}} = -\frac{e_k^\top A_{\mathcal{B}}^{-1} b}{e_k^\top A_{\mathcal{B}}^{-1} \bar{b}}. \tag{A.3}$$

Substitute the definition of $\lambda^*$ from (A.3) into (A.2), we notice that the expression in (A.2) is $\mathbf{0}$. Since $A_{\mathcal{B}}$ is invertible, we have $x_{\mathcal{B}}^* + \lambda^* \bar{x}_{\mathcal{B}} = x_{\mathcal{B}^*}^* + \lambda^* \bar{x}_{\mathcal{B}^*}$, and thus the new dictionary is still optimal at $\lambda^*$.

Case II. When calculating $\lambda^*$ given by (3.10), if, on the other hand, the constraint corresponds to an index $j \in \mathcal{N}$, then $x_{\mathcal{B}^*}^* + \lambda^* \bar{x}_{\mathcal{B}} \geq \mathbf{0}$ is guaranteed by the way we choose leaving variable. It remains to show that it is still dual feasible.

Again, we observe that $A_{\mathcal{B}^*}$ is obtained by changing one column of $A_{\mathcal{B}}$ (say, $a_i$) to another column vector from $A_{\mathcal{N}}$ (say, $a_j$), and we denote $u = a_j - a_i$ as the difference of these two vectors. Without loss of generality, we assume the replacement occurs at the $k$-th column of $A_{\mathcal{B}}$. Sherman-Morrison formula gives

$$A_{\mathcal{B}^*}^{-1} A_{\mathcal{B}} = I - \frac{A_{\mathcal{B}}^{-1} ue_k^\top}{1 + e_k^\top A_{\mathcal{B}}^{-1} u} = I - \frac{pe_k^\top}{1 + e_k^\top p} = \begin{pmatrix} 1 & & -\frac{p_1}{1+p_k} & & \\ & \ddots & \vdots & & \\ & & \frac{1}{1+p_k} & & \\ & & \vdots & \ddots & \\ & & -\frac{p_m}{1+p_k} & & 1 \end{pmatrix}, \tag{A.4}$$

where $p = A_{\mathcal{B}}^{-1} u$, and $p_l$ denotes the $l$-th entry of $p$. Observe that in (A.4), only the $k$-th column is different from the identity matrix.

Dual feasible requires that $z_{\mathcal{N}}^* = (A_{\mathcal{B}}^{-1} A_{\mathcal{N}})^\top c_{\mathcal{B}} - c_{\mathcal{N}} \geq \mathbf{0}$. Since $(A_{\mathcal{B}}^{-1} A_{\mathcal{B}})^\top c_{\mathcal{B}} - c_{\mathcal{B}} = \mathbf{0}$, we slightly change the dual feasible condition to: $(A_{\mathcal{B}}^{-1} A)^\top c_{\mathcal{B}} - c \geq \mathbf{0}$. In the parametric linear programming sense, $c \leftarrow c + \lambda \bar{c}$ and $c_{\mathcal{B}} \leftarrow c_{\mathcal{B}} + \lambda \bar{c}_{\mathcal{B}}$. We only need to show that $(A_{\mathcal{B}}^{-1} A)^\top (c_{\mathcal{B}} + \lambda^* \bar{c}_{\mathcal{B}}) - (c + \lambda^* \bar{c}) = (A_{\mathcal{B}^*}^{-1} A)^\top (c_{\mathcal{B}^*} + \lambda^* \bar{c}_{\mathcal{B}^*}) - (c + \lambda^* \bar{c})$. Consider the following term:

$$\begin{aligned}
&(A_{\mathcal{B}^*}^{-1} A)^\top c_{\mathcal{B}^*} - c + \lambda^*[(A_{\mathcal{B}^*}^{-1} A)^\top \bar{c}_{\mathcal{B}^*} - \bar{c}] - \{(A_{\mathcal{B}}^{-1} A)^\top c_{\mathcal{B}} - c + \lambda^*[(A_{\mathcal{B}}^{-1} A)^\top \bar{c}_{\mathcal{B}} - \bar{c}]\} \\
=&A^\top (A_{\mathcal{B}^*}^{-1})^\top (c_{\mathcal{B}^*} + \lambda^* \bar{c}_{\mathcal{B}^*}) - A^\top (A_{\mathcal{B}}^{-1})^\top (c_{\mathcal{B}} + \lambda^* \bar{c}_{\mathcal{B}}) \\
=&A^\top (A_{\mathcal{B}}^{-1})^\top (A_{\mathcal{B}^*}^{-1} A_{\mathcal{B}})^\top (c_{\mathcal{B}^*} + \lambda^* \bar{c}_{\mathcal{B}^*}) - A^\top (A_{\mathcal{B}}^{-1})^\top (c_{\mathcal{B}} + \lambda^* \bar{c}_{\mathcal{B}}) \\
=&-\alpha A^\top (A_{\mathcal{B}^*}^{-1})^\top e_k,
\end{aligned} \tag{A.5}$$

where $\alpha$ is a constant. According to (A.4), we have

$$
\begin{aligned}
\alpha &= \sum_{l \in \mathcal{B}^* \setminus j} \frac{(c_l + \lambda^* \bar{c}_l) p_l}{1 + p_k} - \frac{c_j + \lambda^* \bar{c}_j}{1 + p_k} + c_i + \lambda^* \bar{c}_i \\
&= \sum_{l \in \mathcal{B}} \frac{(c_l + \lambda^* \bar{c}_l) p_l}{1 + p_k} + \frac{c_i + \lambda^* \bar{c}_i - c_j - \lambda^* \bar{c}_j}{1 + p_k} \\
&= \frac{(c_\mathcal{B} + \lambda^* \bar{c}_\mathcal{B})^\top A_\mathcal{B}^{-1} u + c_i + \lambda^* \bar{c}_i - c_j - \lambda^* \bar{c}_j}{1 + p_k} \\
&= \frac{(c_\mathcal{B} + \lambda^* \bar{c}_\mathcal{B})^\top A_\mathcal{B}^{-1} (a_j - a_i) + c_i + \lambda^* \bar{c}_i - c_j - \lambda^* \bar{c}_j}{1 + p_k} \\
&= \frac{(c_\mathcal{B} + \lambda^* \bar{c}_\mathcal{B})^\top (A_\mathcal{B}^{-1} a_j - e_k) + c_i + \lambda^* \bar{c}_i - c_j - \lambda^* \bar{c}_j}{1 + p_k} \qquad \text{since } A_\mathcal{B}^{-1} a_i = e_k \\
&= \frac{(c_\mathcal{B} + \lambda^* \bar{c}_\mathcal{B})^\top (A_\mathcal{B}^{-1} a_j) - c_j - \lambda^* \bar{c}_j}{1 + p_k} \\
&= \frac{(c_\mathcal{B}^\top A_\mathcal{B}^{-1} a_j - c_j) + \lambda^* (\bar{c}_\mathcal{B}^\top A_\mathcal{B}^{-1} a_j - \bar{c}_j)}{1 + p_k},
\end{aligned}
\tag{A.6}
$$

where $c_i$ and $c_j$ are the entries in $c$, with indices corresponding to $a_i$ and $a_j$, and $\bar{c}_i$ and $\bar{c}_j$ are the entries in $\bar{c}$ and defined similarly .

Recall in this case

$$
\lambda^* = \max_{j \in \mathcal{N}, \bar{z}_{\mathcal{N}_j} > 0} -\frac{z_{\mathcal{N}_j}^*}{\bar{z}_{\mathcal{N}_j}} = -\frac{(A_\mathcal{B}^{-1} a_j)^\top c_\mathcal{B} - c_j}{(A_\mathcal{B}^{-1} a_j)^\top \bar{c}_\mathcal{B} - \bar{c}_j}.
\tag{A.7}
$$

Substitute the definition of $\lambda^*$ from (A.7) into (A.6), we observe that $\alpha = 0$ and thus the dual feasible is guaranteed in the new dictionary. This proves Theorem 3.1. $\qquad\square$

# B    Proof of Theorem 3.5

For notational simplicity, we omit the superscript $\lambda'$ in $\widehat{\mu}$ Before we proceed with the statistical properties of the Dantzig selector, we first introduce the following lemmas.

**Lemma B.1** (Bühlmann and Van De Geer (2011)). Suppose that Assumptions 3.2 and 3.4 hold. Define $\widehat{\Delta} = \widehat{\theta} - \theta^*$. We have

$$
\|\widehat{\Delta}_{\overline{\mathcal{S}}}\|_1 \le \|\widehat{\Delta}_{\mathcal{S}}\|_1.
\tag{B.1}
$$

Moreover, we have

$$
\min_{\|\Delta_{\overline{\mathcal{S}}}\|_1 \le \|\Delta_{\mathcal{S}}\|_1} \frac{\Delta^T \nabla^2 \mathcal{L}(\theta) \Delta}{\|\Delta\|_2^2} \ge \frac{\rho_-(s^* + 2\widetilde{s})}{4}
\tag{B.2}
$$

The proof of Lemma B.1 is provided in Bühlmann and Van De Geer (2011), and therefore is omitted. Note that (B.1) in Lemma B.1 implies that $\widehat{\theta}$ lies in a restricted cone-shape set, and (B.2) implies that (B.1) combined with Assumption 3.4 implies the restricted eigenvalue condition. The next lemma presents the statistical rates of convergence of the Dantzig selector.

**Lemma B.2** (Candes and Tao (2007)). Suppose that Assumptions 3.2 and 3.4 hold. We have

$$
\|\Delta\|_2 = \frac{C_1 \sqrt{s^*} \lambda}{\rho_-(s^* + 2\widetilde{s})} \quad \text{and} \quad \|\Delta\|_1 = \frac{C_2 s^* \lambda}{\rho_-(s^* + 2\widetilde{s})}
\tag{B.3}
$$

The proof of Lemma B.2 is provided in Candes and Tao (2007), and therefore is omitted. Based on Lemmas B.1 and B.2, we can further characterize the statistical properties of $\nabla \mathcal{L}(\widehat{\theta})$ in the following lemma.

**Lemma B.3.** Suppose that Assumptions 3.2 and 3.4 hold. We have

$$
\left| \left\{ j \mid |\nabla_j \mathcal{L}(\widehat{\theta})| \ge \frac{3\lambda}{4}, \ j \in \overline{\mathcal{S}} \right\} \right| \le \widetilde{s}
\tag{B.4}
$$

*Proof.* By Assumption 3.2, we have $\lambda \geq 8\|\nabla\mathcal{L}(\theta^*)\|_\infty$, which further implies

$$\left|\left\{j \mid |\nabla_j\mathcal{L}(\theta^*)| \geq \lambda/8,\ j \in \overline{\mathcal{S}}\right\}\right| = 0. \tag{B.5}$$

We then consider an arbitrary set $\mathcal{S}'$ such that

$$\mathcal{S}' = \left\{j \mid |\nabla_j\mathcal{L}(\widehat{\theta}) - \nabla_j\mathcal{L}(\theta^*)| \geq 5\lambda/8,\ j \in \overline{\mathcal{S}}\right\}.$$

Let $s' = |\mathcal{S}'|$. Then there exists $v$ such that

$$\|v\|_\infty = 1,\quad \|v\|_0 \leq s',\quad \text{and}\quad 5s'\lambda/8 \leq v^\top(\nabla\mathcal{L}(\widehat{\theta}) - \nabla\mathcal{L}(\theta^*)).$$

Since $\mathcal{L}(\widehat{\theta})$ is twice differentiable, then by the mean value theorem, there exists some $z_1 \in [0,1]$ such that

$$\ddot{\theta} = z_1\theta + (1-z_1)\theta^* \quad \text{and}\quad \nabla\mathcal{L}(\widehat{\theta}) - \nabla\mathcal{L}(\theta^*) = \nabla^2\mathcal{L}(\ddot{\theta})\Delta.$$

Then we have

$$\frac{5s'\lambda}{8} \leq v^\top\nabla^2\mathcal{L}(\ddot{\theta})\Delta \leq \sqrt{v^\top\nabla^2\mathcal{L}(\ddot{\theta})v}\sqrt{\Delta^\top\nabla^2\mathcal{L}(\ddot{\theta})\Delta}.$$

Since we have $\|v\|_0 \leq s'$, then we obtain

$$\frac{3s'\lambda}{4} \leq \sqrt{\rho_+(s')}\sqrt{s'}\sqrt{\Delta^\top(\nabla\mathcal{L}(\widehat{\theta}) - \nabla\mathcal{L}(\theta^*))}$$

$$\leq \sqrt{\rho_+(s')}\sqrt{s'}\sqrt{\|\Delta\|_1 \cdot \|\nabla\mathcal{L}(\widehat{\theta}) - \nabla\mathcal{L}(\theta^*)\|_\infty}$$

$$\leq \sqrt{\rho_+(s')}\sqrt{s'}\sqrt{\|\Delta\|_1(\|\nabla\mathcal{L}(\widehat{\theta})\|_\infty + \|\nabla\mathcal{L}(\theta^*)\|_\infty)}$$

$$\leq \sqrt{\rho_+(s')}\sqrt{s'}\sqrt{\|\Delta\|_1(\|\nabla\mathcal{L}(\widehat{\theta}) - \lambda\xi\|_\infty + \lambda\|\widetilde{\xi}\|_\infty + \|\nabla\mathcal{L}(\theta^*)\|_\infty)}$$

$$\leq \sqrt{\rho_+(s')}\sqrt{s'}\sqrt{\frac{115s^*\lambda^2}{12\rho_-(s^* + \widetilde{s})}}.$$

By simple manipulation, we have

$$\frac{5\sqrt{s'}}{8} \leq \sqrt{\rho_+(s')}\sqrt{\frac{115s^*}{12\rho_-(s^* + \widetilde{s})}},$$

which implies

$$s' \leq \frac{184\rho_+(s')}{15\rho_-(s^* + \widetilde{s})} \cdot s^*.$$

Since $s' = |S'|$ attains the maximum value such that $s' \leq \widetilde{s}$ for arbitrary defined subset $\mathcal{S}'$, we obtain $s' \leq \widetilde{s}$. Then by simple manipulation, we have

$$\left|\left\{j \mid |\nabla_j\mathcal{L}(\widehat{\theta}) - \nabla_j\mathcal{L}(\theta^*)| \geq 5\lambda/8,\ j \in \overline{\mathcal{S}}\right\}\right| \leq 13\kappa s^* < \widetilde{s}. \tag{B.6}$$

Thus, (B.5) and (B.6) imply

$$\left|\left\{j \mid |\nabla_j\mathcal{L}(\widehat{\theta})| \geq 3\lambda/4,\ j \in \overline{\mathcal{S}}\right\}\right| \leq \widetilde{s}.$$

$\square$

By the complementary slackness, we have $\widehat{\mu}_j(\nabla_j\mathcal{L}(\widehat{\theta}) - \lambda) = 0$ and $\widehat{\gamma}_j(-\nabla_j\mathcal{L}(\widehat{\theta}) - \lambda) = 0$. By (B.3), we know

$$\left|\left\{j \mid \widehat{\mu}_j \neq 0 \text{ or } \widehat{\gamma}_j \neq 0,\ j \in \overline{\mathcal{S}}\right\}\right| \leq \widetilde{s}. \tag{B.7}$$

Thus, we show that the optimal dual variables are sparse. The cardinality is at most $2s^* + \widetilde{s}$.

To control the sparsity of the primal variables, we directly use the following lemma.

**Lemma B.4** (Gai et al. (2013)). Suppose that Assumptions 3.2 and 3.4 hold. Given the design matrix satisfying

$$\|X_{\overline{\mathcal{S}}}^\top X_{\mathcal{S}}(X_{\mathcal{S}}^\top X_{\mathcal{S}})^{-1}\|_\infty \leq 1 - \zeta,$$

where $\zeta > 0$ is a generic constant, we have $\widehat{\theta}_j = 0$ for any $j \in \overline{\mathcal{S}}$.

The proof of Lemma (B.4) is provided in Gai et al. (2013). Lemma B.4 guarnatees that $\widehat{\theta}$ does not select any irrelevant coordinates. Thus, we complete the proof.