[Reviews · NeurIPS 2017]

Reviewer 1



This paper extends simplex algorithm to several sparse learning problem with regularization parameter. The proposed method can collect all the solutions (corresponding to different values of the regularization parameter) in the process of simplex algorithm. It is an efficient way to get the sparse solution path and avoid tuning the regularization parameter. The connection between path Dantzig selector formulation and sensitivity analysis looks interesting to me. Major comments: - The method used in this paper seems closely related to the sensitivity analysis of LP. What is the key difference? It looks like just an application of sensitivity analysis. - The paper mentioned that the number of iterations is linear in the number of nonzero variables empirically. Is there any guarantee for such linear dependency? Experiment: - Table 1 is not necessary since PSM never violate the constraints. - Simplex method is not very efficient for large scale LP. I did not see comparison with interior based approaches or primal dual approaches. Authors are expected to make such comparison to experiments more convincing. Typo/question: line 122: z^+, z^- => t^+, t^- line 291: solutio => solution line 195: what is the relation between the feasibility condition and the sign of b and c? line 201: \bar{x}_B and \bar{z}_N are always positive? (otherwise how to always guarantee the feasibility when \lambda is large) line 207: here we see an upper-bound for \lambda, it contradicts the claim that the feasibility will be guaranteed when \lambda is large enough. In addition, is it possible that \lambda_min > \lambda_max? line 236: can we guarantee that there is no cycle in such searching?

Reviewer 2



This paper exposes the use of a modified simplex algorithm, the parametric simplex method, to solve sparsity-related problems such as the Dantzig selector, Sparse SVM or again differential network estimation, over a range of regularization parameters. Along with some experiments, this work provides the theoretical analysis of the algorithm concerning its correctness and the sparsity of the iterates. I report some of my concerns below: - My main concern is relative to the experiments, where only very sparse vectors are considered. This raises the question of how the method performs when the sparsity is not that low. - In the context of machine learning, I would like to see how the method performs against coordinate descent-like algorithms (for instance on the Dantzig selector). - Does the fact that the Basis Pursuit, which is also a linear program, is not included is because the authors believes there is enough work on it, or is it because the algorithm does not work well in this case ? It seems to me that BP is the standard linear program to check ... On the bright side, the paper is clearly written, the supplementary materials is reasonably long (3 pages), and everything is technically sounds.

Reviewer 3



In this paper, the authors propose a parametric simplex method for sparse learning. Several advantages over other competing methods are discussed in detail. More numerical experiments are provided to prove the efficiency of the proposed algorithm. The authors claim that first order method work well only for moderate scale problems, but in the experiments, I do not think the author considered large scale case. As a result I suggest the authors to further modify and calculate large-scale case. My main concern for this paper is that the simplex method is just a classical method for linear programming. The authors have not carefully discuss their main contributions the parametric type modification and the novelty.